# Comparison of Two Field Deployable PCR Platforms for SARS-CoV-2 and Influenza A and B Viruses’ Detection [note 1]

**DOI:** 10.3390/pathogens14010027

**Published:** 2025-01-03

**Authors:** Hakim Bouamar, Grace M. Reed, Wanda Lyon, Hector Lopez, Anna Ochoa, Susana N. Asin

**Affiliations:** 1Center for Advanced Molecular Detection, 59th Medical Wing/Science & Technology, Joint Base San Antonio, Lackland, TX 78236, USAhector.lopez184.ctr@health.mil (H.L.); 2711th Human Performance Wing, Airman Systems Directorate, Air Force Research Laboratory, Wright-Patterson AFB, Dayton, OH 45433, USA; wanda.lyon@us.af.mil; 3US Army FUTURE COMMANDS, Fort Sam Houston, San Antonio, TX 78234, USA; anna.r.ochoa.ctr@health.mil

**Keywords:** SARS-CoV-2, influenza A and B, point-of-care diagnostic test, Biomeme, MolBio

## Abstract

Background: Respiratory viral infections are a major public health challenge and the most diagnosed medical condition, particularly for individuals living in close proximity, like military personnel. We compared the sensitivity and specificity of the Biomeme Franklin^TM^ and Truelab^®^ RT-PCR thermocyclers to determine which platform is more sensitive and specific at detecting SARS-CoV-2 and influenza A and B viruses. Methodology: RNA extracted from nasopharyngeal swabs of infected and uninfected individuals was tested on the Biomeme Franklin^TM^ at Lackland and the Truelab^®^ at Wright Patterson Air Force bases. Results: We found an 88% and 71% positivity rate in SARS-CoV-2-infected samples tested on Biomeme and Truelab^®^, respectively. Likewise, we found a 49% and 80% positivity rate in influenza-positive samples tested on Biomeme and Truelab^®^, respectively. One hundred percent of uninfected swab samples tested negative for SARS-CoV-2 on both platforms. Conversely, 91% and 100% of uninfected swabs tested negative for flu on Biomeme and Truelab^®^, respectively. Significance: Differences in specificity and sensitivity in detection of SARS-CoV-2 and influenza between Biomeme and Truelab^®^ suggest that Truelab^®^ is a more promising and potentially deployable diagnostic platform for SARS-CoV-2 and influenza viruses’ detection in an austere environment.

## 1. Introduction

Acute respiratory infections (ARIs) rank fourth on the global causes of death, and lower respiratory tract infections remain the world’s most deadly communicable diseases, despite a remarkable reduction in disease burden and mortality [1]. ARIs are one of the main causes of morbidity in the world, particularly in children under 5 years of age and adults over 65 years old [2]. Likewise, ARIs are the most diagnosed medical condition among recruit trainees, posing a significant threat to the operational effectiveness of our troops [3]. Compared to civilians, military service members (SMs) are at a higher risk for ARIs likely due to crowded living conditions, harsh working environment, physical and psychological stress, and exposure to emergent respiratory pathogens in disease endemic areas [4,5]. Military SMs are also at increased risk for ARIs-associated hospitalizations, mainly pneumonia resulting from increased infections with influenza viruses, coronaviruses, and other viruses [6,7,8]. Department of Defense (DoD) Global Respiratory Pathogen Surveillance Program data from 104 sites across the globe identified ARIs among the most frequent respiratory pathogens infecting clinical samples throughout the 2021–2022 surveillance season [9]. During this period, a total of 65,475 respiratory specimens were tested, among which 26,794 (41%) specimens tested positive for respiratory pathogens. About 61% of the specimens came from outside the continental US (OCONUS), 22% were from the Western U.S., and 17% came from the Eastern U.S. SARS-CoV-2 (70.9%) and Respiratory Syncytial Virus (58.0%) were most detected at OCONUS sites, while influenza (45.0%) and rhinovirus/enterovirus (41.7%) were most detected in the Eastern U.S. Other pathogens (39.6%), such as adenovirus, seasonal coronavirus, human bocavirus, human metapneumovirus, and parainfluenza, were most frequently detected in the Western U.S.

Respiratory pathogens of major military concern include different strains of influenza viruses such as those that caused the 1918–1919 pandemic, 2009 H1N1 pandemic, and annual seasonal influenza viral strains [7,10]. Although most people recover from influenza within a week without requiring medical attention, the virus can still worsen symptoms of other chronic respiratory and non-respiratory conditions including sepsis, occasionally resulting in death [11]. Moreover, a study conducted in the southern hemisphere demonstrated the flu vaccine to be only 52% effective, likely explaining the 15% hospitalization rate among vaccinated civilians with severe ARI [12].

Human coronaviruses are an additional biological threat to military SM. Coronaviruses are a group of RNA viruses with several strains causing severe acute respiratory distress syndrome (SARS). The novel coronavirus 2 (SARS-CoV-2) caused a coronavirus disease (COVID)-19 pandemic at the end of 2019 and to date has claimed nearly seven million lives worldwide, leading to immeasurable damage to the global economy. Hospitalized COVID-19 patients suffered from acute hypoxemic respiratory failure [13], which was the primary attributable cause of death due to multiple organ failure and severe inflammatory response [14,15,16]. Many countries, including the United States, were caught off-guard by the rate of SARS-CoV-2 transmission and patient deaths, with death occurring within between 5.7 and 19 days post-infection [17,18].

Thus, timely and accurate identification of respiratory pathogens of military relevance is essential for the early implementation of mitigation and therapeutic strategies known to be more effective at reducing pathogen transmission in austere environments. These public health strategies are expected to result in fewer lost duty days and enhanced military operational readiness.

Due to its high sensitivity and specificity, Reverse Transcriptase-Polymerase Chain Reaction (RT-PCR) is considered the gold standard for the detection of SARS-CoV-2 and influenza A and B viruses in upper respiratory tract samples. These testing methodologies require technically skilled personnel and can be time-consuming. As an alternative to these delayed sample-to-results times, several portable point of care diagnostic platforms are being developed. Herein, we evaluated the Biomeme Franklin^TM^ RT-PCR thermocycler (Biomeme, Philadelphia, PA, USA) and the Truelab^®^ RT PCR thermocycler by MolBio (MolBio Diagnostics, Verna Industrial Estate, Verna, India), two potential field-deployable, nucleic acid amplification-based diagnostic platforms for sensitivity and specificity on SARS-CoV-2 and influenza A/B detection.

The Biomeme Franklin^TM^ system is a lightweight, portable, battery-operated quantitative PCR device that can test biological samples without the use of centrifugation, frozen reagents, or a power source. This system can detect up to 27 individual targets or 9 targets in triplicate [19].

The Truenat COVID-19 and influenza A/B assays are molecular tests that run on the portable, fully automated, battery-operated Truelab^®^ platform [20,21]. The Truelab^®^ platform is a chip-based real-time reverse transcription duplex Taqman PCR test that offers “sample-to-result” capability in less than an hour and can perform up to four tests per run. Truenat COVID-19 detects the *Envelope* (*E*) and *open reading frame* (*Orf*)1a genes, while Truenat influenza A and B amplify the *membrane protein* (*M*) gene and *non-structural protein* (*NSP*) gene, respectively. The PCR reagents are lyophilized and can be stored at room temperature.

In 2020, the World Health Organization (WHO) urged the scientific community to increase and improve the development of nucleic acid amplification testing platforms, underscoring the need for simplicity and portability [22]. Following WHO recommendations, the main goal of this study was to compare the sensitivity and specificity of the Biomeme Franklin^TM^ and Truenat RT-PCR assays for SARS-CoV-2 and influenza A/B detection in nasopharyngeal (NP) swabs procured from infected and uninfected civilians. Testing results from Clinical Laboratory Improvement Amendments (CLIA)-certified laboratories at the time of sample collection were used as a reference standard.

## 2. Materials and Methods

### 2.1. Clinical Samples

This study received a “not research determination” by the 59th Medical Wing Institutional Review Board (Protocol # FWH2021113N). A total of 394 NP swabs were collected from civilians who tested positive for either SARS-CoV-2 or influenza A/B viruses, as well as from uninfected patients. Patients were recruited and consent was obtained by iSpecimen (Lexington, MA, USA) under an Institutional Review Board-approved protocol. NP swabs were collected in viral transport medium (VTM) (Thermo Scientific, Waltham, MA, USA) on the same day as being tested by the CLIA-certified laboratory, then frozen and shipped to the Center for Advanced Molecular Detection (CAMD, Lackland, SA, TX, USA). The data set provided by iSpecimen was limited to age, gender, race, ethnicity, and type of molecular and serological diagnostics platform. The overview of the study design is shown in Figure 1.

### 2.2. Ribonucleic Acid (RNA) Extraction

RNA was extracted from 200 µL of NP swabs using King Fisher (Thermo Scientific, Waltham, MA, USA) following the manufacturer’s instructions. Aliquots of the same extracted RNA were tested for the presence or absence of SARS-CoV-2 and influenza A/B viruses on Biomeme Franklin^TM^ at the CAMD and on the Truelab^®^ by MolBio Diagnostics at the 711th Human Performance Wing, respectively. This approach allowed for direct comparison of the RNA isolated from the same NPs. Given that Biomeme Franklin^TM^ and the Truelab^®^ are experimental assays, CLIA testing results by the clinical sites at the time of sample collection were used as a gold standard.

### 2.3. Limit of Detection (LoD) Experiments

The LoD is defined as the lowest viral concentration at which 95% of replicates tested positive for either SARS-CoV-2 or influenza A/B. LoD experiments for SARS-CoV-2 were conducted on the Truelab^®^ and Cepheid platforms using six 1:2 serial dilutions of Heat-Inactivated 2019 Novel Coronavirus (ATCC^®^ VR-1986HK™; Manassas, VA, USA). Serial dilutions were made from a stock concentration of 4.2 × 10^8^ genome copies/mL. Likewise, 1:2 serial dilutions from a stock concentration of 2.5 × 10^5^ genome copies/mL of influenza A virus (H1N1, ATCC^®^ VR-95DQ^TM^, Manassas, VA, USA) were made to define the sensitivity of influenza virus detection. The FDA-approved Cepheid^®^ GeneXpert^®^ Xpress plus system (Cepheid^®^, Sunnyvale, CA, USA) was used as a gold standard. This assay is highly specific and sensitive for SARS-CoV-2 and influenza A and B detection. The Xpert SARS-CoV-2/flu/RSV plus assay targets the *nucleocapsid* (*N2*) gene, the *envelope protein* (*E*) gene of SARS-CoV-2, *RNA-dependent RNA polymerase* (*RdRP*), and the human *SPC* gene as an internal control. The result is positive if at least one of the three viral genes is amplified as the instrument reads all viral target genes in a single channel without differentiation between gene targets. The LoD test performed on the Xpert SARS-CoV-2/flu/RSV plus assay for SARS-CoV-2 was reported by Cepheid, the manufacturer, and other laboratories to be ~138–200 viral copies/mL [23]. Truenat COVID-19 detects the *E* and *Orf1a* genes of SARS-CoV-2 and the human *Rnase P* gene as an internal control. The Truelab^®^ platform displays the viral load for positive specimens as a range of Ct values and assigns the result “high” for Ct < 20, “medium” for 20 < Ct < 25, low for 25 < Ct < 30, and very low for Ct > 30. The result is considered positive only if *Orf1a* is detected, while the detection of *E* gene expression alone is interpreted as SARS-CoV-2 presumptive positive. MolBio claims a sensitivity and specificity of 100%, with lower limit of detection at 480 and 487 copies per mL for *Orf1a* and the *E* gene, respectively [24]. As for influenza A and B, the Xpert SARS-CoV-2/flu/RSV plus assay amplifies the genes encoding the *polymerase basic 2* (*PB2*) and *acidic* (*PA*) subunits of the RNA polymerase complex of influenza A and the genes encoding the matrix protein and nonstructural protein of influenza B. Cepheid has reported a LoD for flu A of 0.007 TCID_50_/mL if either of the two genes are detected.

The SARS-CoV-2 LoD on Biomeme Franklin^TM^ was previously published [19]. No LoD experiments were conducted for influenza A/B on the Biomeme Franklin^TM^ system as the company discontinued the product and did not provide their data when evaluating the sensitivity of this assay.

### 2.4. Platform Comparisons

The Biomeme Franklin^TM^ and Truelab^®^ Micro PCR diagnostic platforms were compared for sensitivity and specificity using RNA extracted from 102 SARS-CoV-2-positive samples, 236 influenza A/B-positive samples, and 56 negative samples. Data were normalized to the CLIA testing result at the time of sample collection. All samples were tested following the manufacturer’s instructions. Classification of a positive or negative result was based solely on the platform being used. Positive percent agreement (PPA) and negative percent agreement (NPA) for either SARS-CoV-2 or influenza virus pathogen were calculated based on both the CLIA lab assay and either Truenat or Biomeme assays yielding the same positive or negative result over the total number of samples tested.

### 2.5. Statistical Analysis

Statistical analyses were performed using R version 4.2.1 (2022) and the R package ‘jtools’ (Free Software under the terms of the Free Software Foundation’s GNU General Public License). Sensitivity and specificity for each system were determined with standard calculations and normalized to the CLIA test result. We used Cohen’s kappa statistics to estimate agreement among testing platforms and test the null hypothesis that agreement was random. Kappa statistics ranges between 0 and 1, whereas 0 denotes no agreement and 1 denotes perfect agreement. Likewise, values ranging 0.01–0.20 denote slight agreement, 0.21–0.40 denote fair agreement, 0.41–0.60 denote moderate agreement, 0.61–0.80 denote substantial agreement, and 0.81–1.00 denote almost perfect agreement [25]. We used Pearson’s Chi-square test to test the null hypothesis that the platforms are equivalent in terms of sensitivity and specificity [26]. The McNemar’s test *p* value was used to detect the bias effect, which affects the Cohen’s kappa index results [27].

## 3. Results

### 3.1. Flow Chart of Study Design

Figure 1 depicts the experimental design of this study. In brief, upon collection from October 2020 to March 2022, NP swabs were tested for SARS-CoV-2 and influenza A/B at CLIA-certified collection sites. Samples were procured on the same day as CLIA testing, frozen, and shipped to CAMD for RNA extraction. Aliquots of the same patient’s RNAs were tested at CAMD for SARS-CoV-2 and influenza A/B infection on Biomeme Franklin^TM^ and at the 711th Human Performance Wing on the MolBio Truelab^®^.

### 3.2. Participant Demographics

Participants were distributed in three cohorts: the influenza, SARS-CoV-2, and negative cohorts. There were 236 influenza participants, 102 SARS-CoV-2 participants, and 56 negative participants in the cohorts (Figure 1 and Table 1). Within each cohort, data were stratified by age, race, and gender. In general, SARS-CoV-2-infected participants were older than influenza participants, 43 and 14 years of age on average, respectively.

A total of 79% of enrollees were Hispanic, whereas 21% were Caucasians, and both the influenza and SARS-CoV-2 cohorts had similar percentages of female and male participants. The exemption was the control cohort, where 61% of participants were females. Data on race were not reported for influenza-infected samples. In addition, of the 236 influenza-infected participants, there were 39 with missing gender and age data.

### 3.3. CLIA Testing of the NP Swabs at the Time of Sample Collection

Given that molecular testing has become the gold standard for detecting respiratory pathogens because of its high sensitivity and specificity, most of the methods used in CLIA-certified laboratories involve the use of multiplex RT-PCR to diagnose influenza A/B and SARS-CoV-2 [10,28]. As depicted in Table 2, in the influenza cohort, the Cobas system was the most adopted screening platform, followed by Cepheid’s GeneXpert [29,30] and the BioFire Torch. Only 16% of participants were diagnosed with immunoassays, using either the Manual/Meridian or Luminex MagPix kits (Table 2). SARS-CoV-2 and uninfected samples were tested by RT-PCR.

### 3.4. Limit of Detection

The Cepheid^®^ GeneXpert^®^ Xpress plus system demonstrated the greatest sensitivity for SARS-CoV-2 detection compared to the Biomeme Franklin^TM^ and the Truenat assays, with a LoD of 102 viral copies/mL (Figure 2, upper left panel). In contrast, the LoD of Biomeme appeared to be target-dependent. We detected a LoD of 8488 copies per mL and 4576 copies per mL for *Orf1a* and *S* genes, respectively (Figure 2, upper right and lower left panels). All replicates tested positive for SARS-CoV-2 at 4.2 × 10^3^ genomic copies per mL and negative at a concentration of 420 viral genomic copies per ml on the MolBio Truelab^®^ instrument (Figure 2, lower right panel). Thus, the LoD on the Truelab^®^ was estimated to be lower than 4200 copies per mL and greater than 420 copies per mL. Notwithstanding, all three assays detected SARS-CoV-2, with the Cepheid^®^ GeneXpert^®^ SARS-CoV-2/flu/RSV plus assay outperforming Biomeme by an 83- and Truenat by an 18-fold-greater change in sensitivity.

Likewise, Cepheid demonstrated the greatest sensitivity compared to Truenat influenza A/B assay, with a LoD of 36 and 1073 viral copies per ml, respectively (Figure 3). Cepheid outperformed Truenat by a 30-fold-greater change. Thus, the Cepheid demonstrated the greatest sensitivity for flu detection compared to the Truenat influenza A/B assay.

### 3.5. Comparative Testing Between Truenat and Biomeme and CLIA Testing Platforms

The data indicate a 92.4% overall agreement rate for SARS-CoV-2-infected samples tested on the Biomeme and CLIA platforms (Table 3A, left panel). In contrast, samples tested on the Truelab^®^ and CLIA platforms showed an 81.7% overall agreement rate (Table 3B, left panel). Influenza-infected samples demonstrated an overall agreement rate of 57.6% between the Biomeme and CLIA platforms compared to 83.6% between the Truelab^®^ and CLIA platforms, (Table 3A,B, right panels). Sensitivity and specificity calculations for Truelab^®^ and Biomeme platforms were assessed by determining the positive percent agreement (PPA) and negative percent agreement (NPA) relative to the CLIA lab test results [31]. The PPA for SARS-CoV-2 detection was 88% on Biomeme compared to the CLIA platforms and 49.7% for influenza virus detection (Table 3A).

When the same comparison was conducted between samples tested on the Truelab^®^ and CLIA platforms, we detected a PPA of 71.6% and 80.52% for SARS-CoV-2 and influenza, respectively (Table 3B). The negative percent agreement was higher than the positive percent agreement, likely due to the higher specificity of the primers in non-infected samples compared to infected ones. A total of 100% and 91% of samples tested negative on the Biomeme and CLIA platforms for SARS-CoV-2 and influenza, respectively (Table 3A), whereas all samples from uninfected patients tested negative for SARS-CoV-2 and influenza on the Truelab^®^ and CLIA platforms, (Table 3B). Taken together, the Truelab^®^ platform shows a better sensitivity and specificity rate for influenza A detection, whereas Biomeme performs better at detecting SARS-CoV-2.

### 3.6. Agreement Between Diagnostic Platforms

The overall agreement of Biomeme and Truenat with SARS-CoV-2 and influenza CLIA test results was estimated by Cohen’s Kappa statistics. A Kappa statistic of 0.8417 suggested almost-perfect agreement for SARS-CoV-2 detection between Biomeme and CLIA testing platforms (Table 4). In contrast, the agreement was moderate, at 0.5647, between Truenat and CLIA testing platforms. Regarding influenza virus detection, the Kappa statistic indicates fair agreement between Biomeme and CLIA testing platforms, at 0.2311, and substantial agreement, at 0.6409, between Truenat and CLIA testing platforms. The comparison of Biomeme and Truenat demonstrated substantial agreement, at 0.7665, for SARS-CoV-2 detection and fair agreement, at 0.3808, for influenza detection. These data are consistent with the PPA reported in Table 3.

These platforms were not equivalent in sensitivity at detecting either SARS-CoV-2 or influenza virus. The *p* values obtained from McNemar’s test corroborate the agreement tests (Table 4) and strengthen our conclusions.

## 4. Discussion

There are several commercially available FDA-approved molecular devices with POC capability. Multiplex PCR is a validated strategy for the specific and sensitive detection of respiratory viruses. However, these assays require sizeable instruments, multiple steps of biochemical processing, are time-consuming and labor-intensive, and cannot be deployed at the site of pathogen encounter, which prevent them from being used as deployed tools. Furthermore, some molecular diagnostics panels have been approved for a very specific setting such as high-prevalence screening situations and fail when used in a different one. For example, high rates of false positive results for SARS-CoV-2 were observed in a predominantly asymptomatic patient population with the Cobas Liat, which was originally designed for screening symptomatic individuals [32,33,34]. Furthermore, poor analytical sensitivities were documented for isothermal and non-RT-PCR technologies, which led to an increase in clinical false negative results [34,35,36]. Thus, there is an urgent need for more reliable and cost-effective diagnostic platforms that can be used by warfighters in the field to identify ARIs. Ideally, deployable POC diagnostic platforms should be small, use solar or battery power, have lower weight, be more rugged, have lower energy consumption, be simpler to use and faster, and have improved accuracy.

Given that early and accurate identification of respiratory pathogens of military relevance at the point of exposure is relevant to military operational readiness, herein, we evaluated and compared the sensitivity and specificity of Biomeme Franklin^TM^ and the MolBio Truelab^®^ for SARS-CoV-2 and influenza virus detection. Both platforms are potentially deployable and of DoD interest.

Even though Biomeme and MolBio have the potential to enable near-patient diagnostic testing and viral load monitoring with a Ct value, we report significant differences in specificity and sensitivity for SARS-CoV-2 and influenza A/B virus detection between these assays, at least in human NP swabs procured from infected individuals. Furthermore, to be deployable, both platforms must incorporate a module for field RNA extraction.

Overall, the Truelab^®^ platform showed good sensitivity and perfect specificity rate for influenza A detection compared to Biomeme. This was most likely due to Biomeme´s reduced sensitivity in detecting influenza A virus. Likewise, Cohen’s Kappa values and McNemar’s test results indicated significant differences in performance between these platforms, with the Truelab^®^ platform demonstrating higher sensitivity and specificity for influenza A compared to Biomeme Franklin^TM^. Furthermore, the lower amount of viral RNA required by Truelab^®^ for testing (a 10-fold reduction), as well as the lower failure rate per run, i.e., only two invalid runs of 394, suggest that Truelab^®^ would be more reliable as a diagnostic platform compared to Biomeme. One additional advantage of the Truelab^®^ platform is its short runtime (less than one hour), which can decrease the length of a patient’s stay on an emergency ward or expedite intake of antiviral drugs. Taken together, these findings emphasize the relevance of platform-specific considerations for accurate respiratory pathogen detection, especially at the point of pathogen encounter. Additional information on the usability and cost-effectiveness of these platforms is provided in Appendix A.

Besides differences in diagnostic platforms performance, the shipping conditions of biological samples could have reduced the sensitivity of these assays. Even though the NP swabs used in this study were shipped in VTM instead of the recommended manufacturer’s viral lysis media, the manufacturer has reported the Truenat assay to be 100% sensitive and specific with a LoD of 1 × 10^5^ SARS-CoV-2 viral copies per mL when samples were stored in VTM [21]. Thus, differences in performance are unlikely to be explained by sample shipment conditions.

Even though we did not perform near-neighbor testing to define the platform’s specificity, our group previously tested genomic material from nineteen near-neighbor upper respiratory bacterial and viral pathogens, including six strains of other coronaviruses, for cross-reactivity with SARS-CoV-2 on Biomeme [19]. For each near-neighbor pathogen, no targets were detected in the SARS-CoV-2 Go-Strips [19]. Conversely, all near-neighbor upper respiratory pathogens evaluated on the BioFire^®^ RP2.1 used as a gold standard were detected. In addition, Truelab^®^ did not detect influenza B virus when VTM spiked in with RNA from influenza A virus was used, suggesting some degree of specificity of this platform. Furthermore, no cross-reactivity in the performance of the Truenat influenza A/B assay was detected when several microorganisms were evaluated in silico by MolBio (https://www.MolBiodiagnostics.com/index.php, accessed on 11 October 2024).

Of utmost importance, the Truelab^®^ and Biomeme platforms are, at best, near-point-of-care nucleic acid amplification tests because they both require specialized laboratory instruments to isolate RNA. We encourage both companies to integrate an RNA extraction step into their device to promote deployability and limit sample contamination [37].

The major disadvantages of Truelab^®^ is that only up to four samples can be processed per machine at a time, so it is not an ideal instrument for a high-throughput laboratory. Because the Truenat assays used in this study do not allow multiplexing except for influenza A and B, this limitation might increase the number of tests necessary to diagnose a pathogen, especially if the initial test result is negative and needs to be confirmed. Despite this limitation, the Truenat assay has the potential to be used as a pre-screening method to relieve the burden of molecular testing on military hospitals as well as military or private laboratories. The latter is especially true for multiplex assays (multiple pathogen), which is unfortunately not the case for Truenat [38,39].

Respiratory viruses are constantly mutating, and the impact that novel variants of viruses can have on diagnostic platforms’ performance needs to be taken into consideration. Truenat has overcome this issue by detecting two targets for SARS-CoV-2, which increases the probability of detecting the pathogen, even in the scenario of point mutations of one of the two genes. Truenat influenza A detects the slowest-evolving and the more conserve *M* gene of the viral genome [40], thereby preventing false negative results. Thus, it is imperative to constantly validate diagnostic platforms against a panel of variants as viral evolution has been associated with changes in diagnostic test sensitivity, even when targeting gene regions recommended by the CDC and WHO [41,42,43,44]. Furthermore, the ability of a molecular test to detect zoonotic strains with the potential to infect humans would increase their usability in potential emergent pathogens outbreaks such as the swine and avian influenza variants which have been reported to infect human populations [45].

As for Biomeme, the major limitation of this platform is the lack of a company estimation of the LoD for influenza A/B, followed by the discontinuation of the assay. Despite these limitations, our findings provide valuable insights into the strengths and weaknesses of the Biomeme Franklin^TM^ platform.

An additional limitation of this study is that we only tested NP swabs as they are considered the gold standard for clinical testing. We have previously demonstrated the utility of using samples other than NP swabs for SARS-CoV-2 detection, among them saliva, a non-invasive sample that can be self-collected, limiting exposure of health care workers to infection [46]. Although relevant, characterizing the infectivity of the viruses present in these biological samples was beyond the scope of our study.

Ideally, both platforms should have been evaluated in the same military laboratory. However, this study stemmed from a collaboration between two military facilities: the 59th Medical Wing Science and Technology, Center for Advanced Molecular Detection, which was evaluating the Biomeme Franklin^TM^ system towards advanced development, and the 711th Human Performance Wing, which was evaluating MolBio, an additional potential field-deployable PCR platform of DoD interest. The idea was that each agency would leverage their expertise with the respective platform of interest and, using the same samples, would perform a comparison of SARS-CoV-2 and influenza virus detection to identify the optimal PCR platform that the DoD could move forward with. This was a pilot study to select the platform that is likely to perform the best. The next step of this project is to have one laboratory testing both diagnostic platforms using the same samples before conducting a validation trial using prospectively collected clinical samples with the long-term aim of applying for FDA approval.

To evaluate a new diagnostic test for SARS-CoV-2 and influenza A/B, data should be compared to that generated with a gold-standard instrument such as Cepheid. Unfortunately, neither the Cepheid company nor independent laboratories have reported a LoD for influenza A/B that we could use to compare our data against. The Xpert Xpress SARS-CoV-2/flu/RSV assay was discontinued in 2023 and replaced by the Xpert Xpress SARS-CoV-2/flu/RSV Plus, as this contains specific primers targeting more mutations. The analytical performance of the new “Plus” Cepheid assay was drastically improved as a recent publication reported 10 copies/ml and 50 copies/mL for SARS-CoV-2 and flu A and B, respectively [47]. Probit regressions analysis detected a LoD of 102 genome copies/mL and 36 genome copies/mL for SARS-Cov-2 and influenza A, respectively. Thus, our data suggest enhanced sensitivity for influenza A compared to SARS-COV-2 detection, whereas Jensen et al. reported better sensitivity for SARS-CoV-2 compared to influenza A. Notwithstanding, Cepheid outperformed Biomeme Franklin^TM^ and Truenat.

## 5. Conclusions

Overall, our findings suggest that both platforms, Biomeme Franklin^TM^ and Truenat, can detect SARS-CoV-2 and influenza in NP swabs, yet significant improvements are still needed to develop a deployable, rapid, and reliable point-of-care diagnostic test that the DoD can use to accurately identify respiratory pathogens infecting military service members, specifically in austere environments. Thus, definitive conclusions regarding the superiority of one platform over the other cannot be made as further validation with a broader range of samples is needed, and both diagnostic platforms must be evaluated in the same military facility by the same operator. Key areas for improvement include enhancing sensitivity, enabling multiplexing, and integrating field RNA extraction capabilities to ensure these systems meet the operational needs of the DoD.

## Figures and Tables

**Figure 1 pathogens-14-00027-f001:**
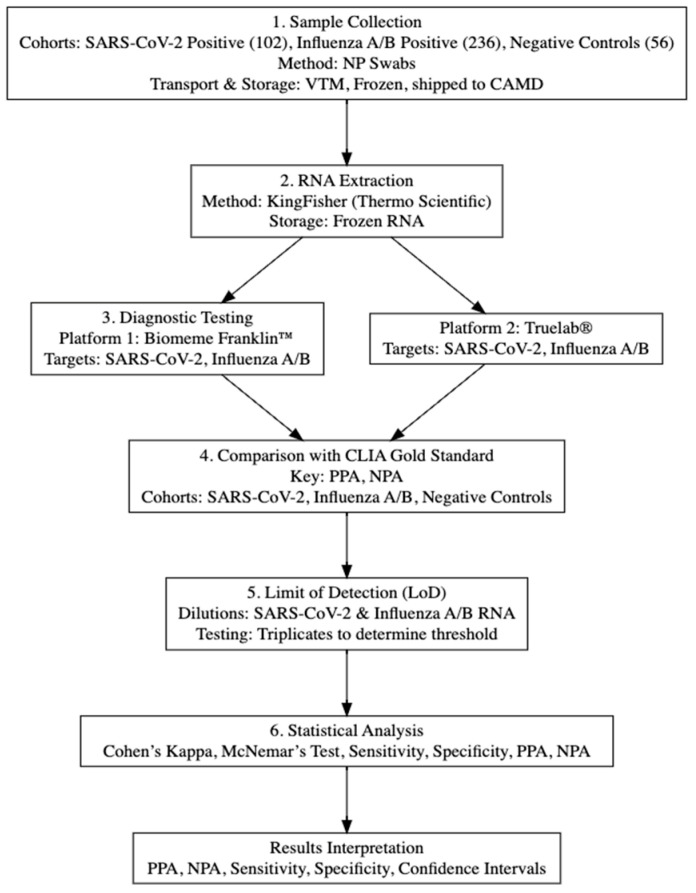
Overview of study design. Participants were assigned to each cohort according to the initial clinical CLIA test results.

**Figure 2 pathogens-14-00027-f002:**
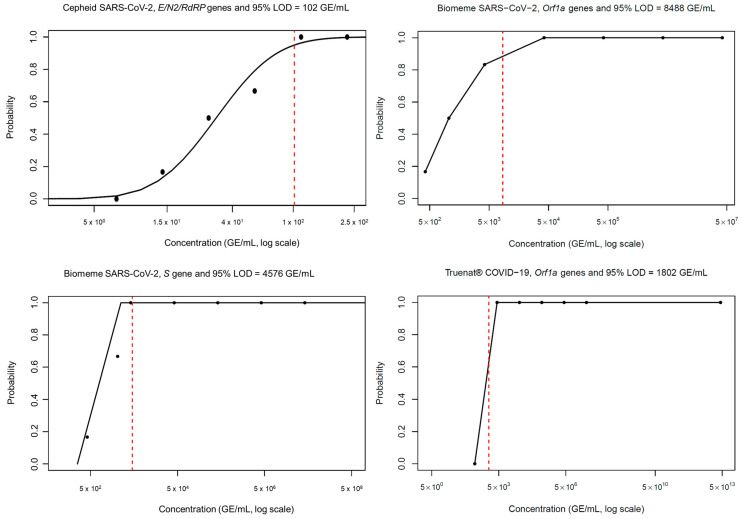
Range of known viral concentrations used to estimate Limit of Detection (LoD). SARS-CoV-2 LoD estimation with the Xpert SARS-CoV-2/flu/RSV plus assay versus the Biomeme Franklin^TM^ assay and Truenat COVID-19 by using serial known dilutions of the heat-inactivated 2019 Novel Coronavirus (ATCC^®^ VR-1986HK). Each data point is the average of six replicates. The red line indicates the concentration at which the assay reaches the threshold for a 95% detection rate.

**Figure 3 pathogens-14-00027-f003:**
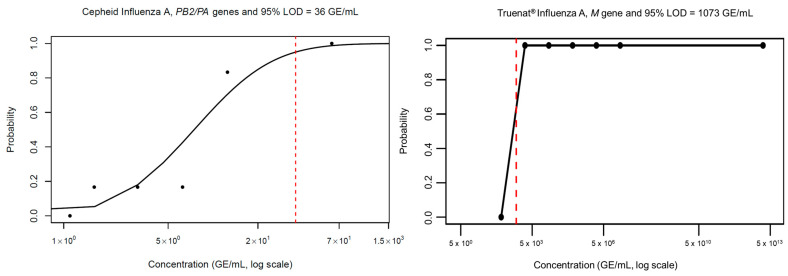
Range of known viral concentrations used to estimate Limit of Detection (LoD). Influenza A limit of detection with the Xpert SARS-CoV-2/flu/RSV plus assay versus the Truenat COVID-19 MolBio using the quantitative genomic RNA from influenza A virus (H1N1) strain A/PR/8/34 (ATCC^®^ VR95DQ™). Each data point is the average of six replicates. The red line indicates the concentration at which the assay reaches the threshold for a 95% detection rate.

**Table 1 pathogens-14-00027-t001:** Participant demographics. The table depicts number of participants in total and in each individual cohort stratified by age, race, and sex.

	**All Patients**	**Influenza Cohort**	**SARS-CoV-2 Cohort**	**Negative Cohort**
**Number of participants**	394	236	102	56
**Age Mean (SD)**	27 (21)	14 (19)	43 (14)	49 (14)
Min	0	0	20	20
Max	94	94	80	71
**Hispanic (%)**	125 (79)	ND	77 (75)	48 (86)
**Caucasian (%)**	33 (21)	ND	25 (25)	8 (14)
**Male (%)**	166 (47)	96 (49)	48 (47)	22 (39)
**Female (%)**	189 (53)	101 (51)	54 (53)	34 (61)

**Table 2 pathogens-14-00027-t002:** Type of multiplex RT-PCR method or immunoassay and platform used to detect SARS-CoV-2 and influenza virus infection at a CLIA-certified laboratory. NP swabs were tested at the time of sample collection, frozen, and shipped to the Center for Advanced Molecular Detection at Lackland Air Force Base.

	Infulenza Cohort	SARS-CoV-2 Cohort	Negative Cohort
**Number of participants diagnosed with Multiplex RT-PCR (%)**	162 (84)	102 (100)	57 (100)
-----Roche Cobas	71 (44)	not provided	not provided
-----Cepheid’s GeneXpert	36 (22)	not provided	not provided
-----Biofire Torch	32 (20)	not provided	not provided
-----Diasorin Integranted Cycler	21 (13)	not provided	not provided
-----Genmark ePlex	2 (1)	not provided	not provided
**Number of participants diagnosed with Immunoassay (%)**	32 (16)	-	-
-----Manual/Meridian kit	23 (72)	-	-
-----Luminex MagPix	9 (28)	-	-

**Table 3 pathogens-14-00027-t003:** Comparison of CLIA testing results to the clinical performance of two sample-to-result molecular assays, Truenat and Biomeme.

**A**		**CLIA Test**					
		**SARS-CoV-2**			**Influenza**		
		**Positive**	**Negative**	**Total**	**Positive**	**Negative**	**Total**
**Biomeme**	**Positive**	90	0	90	115	5	120
	**Negative**	12	56	68	117	51	168
	**Total**	102	56	158	232	56	288
	Overall rates of agreement			92.41%			57.64%
	PPA			88.24%			49.57%
	NPA			100.00%			91.07%
**B**		**CLIA Test**					
		**SARS-CoV-2**			**Influenza**		
		**Positive**	**Negative**	**Total**	**Positive**	**Negative**	**Total**
**MolBio**	**Positive**	73	0	73	186	0	186
	**Negative**	29	56	85	45	43	88
	**Total**	102	56	158	231	43	274
	Overall rates of agreement			81.56%			83.58%
	PPA			71.57%			80.52%
	NPA			100.00%			91.07%

**Table 4 pathogens-14-00027-t004:** Agreement between CLIA testing and diagnostic platforms.

		Cohen’s Kappa (*p*-Value)	McNemar’s Test *p*-Value
**SARS-CoV-2**	Biomeme and CLIA	0.8417 (7.229 × 10^−84^)	0.001496
	Truenat^®^ and CLIA	0.5647 (3.944 × 10^−26^)	5.41 × 10^−11^
	Biomeme and Truenat^®^	0.7665 (9.614 × 10^−55^)	2.42 × 10^−4^
**Influenza A**	Biomeme and CLIA	0.2311 (1.244 × 10^−9^)	2.2 × 10^−16^
	Truenat^®^ and CLIA	0.6409 (4.439 × 10^−30^)	1.999 × 10^−7^
	Biomeme and Truenat^®^	0.3808 (3.686 × 10^−15^)	4.238 × 10^−12^

## Data Availability

The original contributions presented in this study are included in the article/Appendix A. Further inquiries can be directed to the corresponding authors.

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
