# Peer review of "Comparison of Two Field Deployable PCR Platforms for SARS-CoV-2 and Influenza A and B Viruses’ Detection†"

_pathogens, 2025, doi:10.3390/pathogens14010027_

Round 1

Reviewer 1 Report

Comments and Suggestions for Authors

This manuscript evaluates the performance of two field-deployable RT-PCR platforms, aiming to provide recommendations for the rapid diagnosis of respiratory diseases. However, as discussed in the manuscript’s discussion section, this study lacks cross-studies on pathogens related to respiratory diseases other than SARS-CoV-2 and influenza viruses. Nevertheless, for a diagnostic platform based on RT-PCR, the specificity of the kit is of paramount importance, as it directly impacts the ability to make accurate diagnoses and prescribe appropriate medications. Consequently, it is recommended that relevant content be incorporated into the manuscript to address this oversight.

Author Response

This manuscript evaluates the performance of two field-deployable RT-PCR platforms, aiming to provide recommendations for the rapid diagnosis of respiratory diseases. However, as discussed in the manuscript’s discussion section, this study lacks cross-studies on pathogens related to respiratory diseases other than SARS-CoV-2 and influenza viruses. Nevertheless, for a diagnostic platform based on RT-PCR, the specificity of the kit is of paramount importance, as it directly impacts the ability to make accurate diagnoses and prescribe appropriate medications. Consequently, it is recommended that relevant content be incorporated into the manuscript to address this oversight.
Answer: thank you very much for raising this point. The reviewer is correct. Testing the specificity of the diagnostic platform against near-neighbor pathogens is very relevant and was not conducted for this study. However, as stated in lines 341-346 data on near neighbor pathogen testing on the Biomeme was published by our lab in J Clin Virol in 2022. Please see Table 2 of reference #19 entitled “Evaluating sensitivity and specificity of the Biomeme Franklin™ three9 real-time PCR device and SARS-CoV-2 go-strips assay using clinical samples”.
The specificity of the Truenat assay to detect Influenza A/B was assessed by the company against several microorganisms including bacteria and viruses. Please see "Specific performance characteristics" in the Truenat® Influenza A/B brochure.
This information is now included in the discussion section of the manuscript (line 346-351). Near neighbor testing for Influenza on the Truelab® RT-PCR thermocycler. We didn’t detect influenza B virus when viral transport medium spiked in with RNA from influenza A virus was used, suggesting some degree of specificity of this platform (data not shown). 

Reviewer 2 Report

Comments and Suggestions for Authors

This research article titled “Comparison of Two Field Deployable PCR Platforms for Respiratory Viruses Detection” authored by H Bouamar, G Reed, W Lyon, H Lopez, A Ochoa and Susana Asin compares the sensitivity and specificity of the Biomeme FranklinTM and Truelab® RT-PCR thermocyclers to determine which platform is more sensitive and specific at detecting SARS-CoV-2 and Influenza A and B viruses. While the manuscript is well written, experimentation is well carried out and the scientific data analysis is sound, there are a few shortcomings listed below that need to be revised before the manuscript may be considered for publication:

1)    There are a few grammatical errors throughout the text that must be edited before re uploading the manuscript.

2)    The authors are requested to add a workflow/schematic to understand the comparative assay carried out in this research work.

3)    The authors must include a list of other portable RT-PCR thermocyclers available in the market and justify why they chose to specifically compare the Biomeme FranklinTM and Truelab® RT-PCR thermocyclers.

4)    The authors are requested to include a conclusion section where the final concluding statements from the discussion section can be moved.

5)    The authors may add global statistical data on number of ARI cases in military SM in the Introduction section to understand the severity.

6)    The authors are also requested to add a cost analysis and comparison (including reagents if different between the two) between the two thermocycler assays to provide an overall picture of on-field application feasibility.

7)    The COVID pandemic saw the rise of LFA based point of care on-field diagnostic assays approved for initial screening. In the discussion, could the authors mention whether using the current thermocycling assays is better than using the PoC biosensors.

Author Response

1- This research article titled “Comparison of Two Field Deployable PCR Platforms for Respiratory Viruses Detection” authored by H Bouamar, G Reed, W Lyon, H Lopez, A Ochoa and Susana Asin compares the sensitivity and specificity of the Biomeme FranklinTM and Truelab® RT-PCR thermocyclers to determine which platform is more sensitive and specific at detecting SARS-CoV-2 and Influenza A and B viruses. While the manuscript is well written, experimentation is well carried out and the scientific data analysis is sound, there are a few shortcomings listed below that need to be revised before the manuscript may be considered for publication.   

Answer: we thank the reviewer for his/her insightful comments as well as for the careful review of our manuscript. We are glad the reviewer didn’t raise concerns about the experimental design or the data analysis sections of our manuscript. A detailed response addressing each of the Reviewer concerns is depicted below:

2- There are a few grammatical errors throughout the text that must be edited before re uploading the manuscript.

Answer: following the reviewer’s suggestion the re uploaded manuscript has been edited by three native English speakers. Please if the reviewer still identifies grammatical errors and point those to us, we will be very happy to correct them.

3-The authors are requested to add a workflow/schematic to understand the comparative assay carried out in this research work.
Answer: as suggested, we have added a workflow describing the study experimental design to the revised version of the manuscript. Please see New Figure 1 Line 187-190 of re-uploaded manuscript.

4-The authors must include a list of other portable RT-PCR thermocyclers available in the market and justify.
Answer: to the best of our knowledge, we couldn’t find a list of FDA approved portable RT-PCR thermocyclers available in the US market

5- The authors should justify why they chose to specifically compare the Biomeme FranklinTM and Truelab® RT-PCR thermocyclers.
Answer: Both platforms, Biomeme FranklinTM and Truelab® RT-PCR thermocyclers, were of DoD interest as potential deployable.

6- The authors are requested to include a conclusion section where the final concluding statements from the discussion section can be moved.

Answer: following the reviewer’s suggestion we have added a Conclusion paragraph to the manuscript. Please see lines 398-403.

7- The authors may add global statistical data on number of ARI cases in military SM in the Introduction section to understand the severity.
Answer: following the reviewer’s suggestion we have added worldwide statistics on the number of Acute Respiratory Infection cases in Military service members. Please see Introduction Section, lines 47-57.

8- The authors are also requested to add a cost analysis and comparison (including reagents if different between the two) between the two thermocycler assays to provide an overall picture of on-field application feasibility.

Answer: following the reviewer’s suggestion we have added a cost analysis and summary of the field application between Biomeme FranklinTM and Truelab® RT-PCR thermocyclers. Please see Supplementary Table 1.

9-The COVID pandemic saw the rise of LFA based point of care on-field diagnostic assays approved for initial screening. In the discussion, could the authors mention whether using the current thermocycling assays is better than using the PoC biosensors."
FDA approved LFA and Biosensors to detect SARS-CoV-2 and Influenza A/B are based on antigen detection. Therefore, we can’t compare LFA and Biosensors to Biomeme FranklinTM and Truelab® RT-PCR platforms.

Reviewer 3 Report

Comments and Suggestions for Authors

Reviewer Comments

1.       In the Methods section, the volume of the initial sample used for RNA extraction from each samples is unclear. This detail is particularly important, especially considering that the CoV-2 yielded the lowest amount.

2.       In the Results and Discussion section, the authors should clarify whether any viral cultures were attempted and, if so, whether any of these viruses were successfully isolated and characterized.

3.       The potential for the novel viruses found in this study to be recent recombination events of existing coronaviruses should be discussed, as recombination is common and may contribute to the emergence of new coronaviruses.

4.       The authors should redefine the main objective of the study, providing clarification on what is meant by "circulating pathogens." The objective should be more than just a statement; it should be clearly descriptive.

5.       The number of patient samples should also be included in the text.

6.       Two phages were used as quality controls for the extraction process. However, the percentage of recovery after sample processing is not clear. Please include this information.

7.       The manuscript does not specify the time period during which the samples were collected. This information should be clarified.

Author Response

1- In the Methods section, the volume of the initial sample used for RNA extraction from each sample is unclear. This detail is particularly important, especially considering that the CoV-2 yielded the lowest amount.
Answer: following the reviewer’s suggestion the revised version of the manuscript now contains the volume of the initial sample (200 µL) used for RNA extraction. Please see Materials and Methods Section, Line 119.

2- In the Results and Discussion section, the authors should clarify whether any viral cultures were attempted and, if so, whether any of these viruses were successfully isolated and characterized.
Answer: the isolation and characterization of the viruses present in these biological samples was beyond the scope of this manuscript. Following the reviewer’s suggestion we have added this information as an additional limitation of the study. Please see lines 382-384.

3- The potential for the novel viruses found in this study to be recent recombination events of existing coronaviruses should be discussed, as recombination is common and may contribute to the emergence of new coronaviruses.
Answer: the reviewer is correct recombination events could lead to the appearance of novel and potentially more infective variants. However, the diagnostic platforms used in this study only allow the identification of SARS-CoV-2 and Influenza A or B without specific identification of the viral strain or variant of concerns. 

4- The authors should redefine the main objective of the study, providing clarification on what is meant by "circulating pathogens." The objective should be more than just a statement; it should be clearly descriptive.
Answer: following the reviewer’s suggestion, we have clearly described the objective of our study Please see Lines 101-104.

5- The number of patient samples should also be included in the text.
The number of patient samples was reported in the text of the original manuscript. Please see Lines 193-194.

6- Two phages were used as quality controls for the extraction process. However, the percentage of recovery after sample processing is not clear. Please include this information.
Answer: with all due respect we do not understand this question. We didn’t use phages for any extraction process therefore we can’t report on percentage recovery.

7-The manuscript does not specify the time during which the samples were collected. This information should be clarified.
Answer: as suggested we have now specified the time-period of sample collection. Biological specimens were collected between October of 2020 and March of 2022. All samples tested positive or negative for SARS-CoV-2 and Influenza virus at the time of sample collection. The information has been added to the manuscript. Please see lines 181-182.

Reviewer 4 Report

Comments and Suggestions for Authors

Bouamar H. et al. compared the Biomeme FranklinTM and Truelab® RT-PCR thermocyclers for respiratory viruses detection. The manuscript is well-written; however, the viruses were quantified by two independent laboratories, which raises doubts. It would be beneficial to detect respiratory viruses by the scientific personnel of one of the engaged laboratories (to enhance the consistency and reliability of the comparison).

  1. The Xpert SARS-CoV-2/Flu/RSV assay and the examined assay quantified different SARS-CoV-2 genes and internal controls. Estimation of assay efficacy is debatable.
  2. Could the authors explain why they decided to include a third Cepheid platform – the “gold standard” in their study?
  3. Figure 1, the name of SARS-CoV-2 genes should be added to the Cepheid SARS-CoV-2 and Truenat® COVID-19 assays.

Author Response

1- Bouamar H. et al. compared the Biomeme FranklinTM and Truelab® RT-PCR thermocyclers for respiratory viruses’ detection. The manuscript is well-written; however, the viruses were quantified by two independent laboratories, which raises doubts.

Answer: the reviewer’s comment is well taken and was addressed in the discussion lines 386-397, however this study stemmed from a collaboration between two military facilities: the 59th Medical Wing Science and Technology, Center for Advanced Molecular Detection that was evaluating the Biomeme FranklinTM system towards advanced development; and the 711th Human Performance Wing, that was evaluating MolBio an additional potential field-deployable PCR platform of DoD interest. The idea was that each agency will leverage their expertise with the respective platform of interest and using the same samples will perform a comparison of SARS-CoV-2 and Influenza virus detection to identify the optimal PCR platform that the DoD could move forward with.

2- It would be beneficial to detect respiratory viruses by the scientific personnel of one of the engaged laboratories (to enhance the consistency and reliability of the comparison).

Answer: the reviewer is correct; however, this was a pilot study to select the platform that is likely to perform the best. As suggested the next step of this project is to have one laboratory testing both diagnostic platforms using the same samples before conducting a validation trial using prospectively collected clinical samples with the long term of applying for FDA approval.

3- The Xpert SARS-CoV-2/Flu/RSV assay and the examined assay quantified different SARS-CoV-2 genes and internal controls. Estimation of assay efficacy is debatable.

Answer: We believe that our limit of detection (LoD) results as well as findings of the direct comparison of the Biomeme Franklin™ and Truelab® platforms with the CLIA testing gold standard, provide strong evidence supporting the efficacy of these diagnostic platforms. Both diagnostic platforms consistently detected SARS-CoV-2 and Influenza A/B in clinical samples with high sensitivity and across a range of viral concentrations. The data does not suggest any significant concerns regarding the efficacy of the assays, as their performance aligns closely with established standards.

4- Could the authors explain why they decided to include a third Cepheid platform – the “gold standard” in their study?
Answer: The Cepheid GeneXpert® Xpress plus platform is worldwide recognized as the “gold standard” for testing of respiratory pathogens, and it was included in the Limit of Detection (LoD) experiments to provide a reliable benchmark to assess the accuracy and reliability of the new diagnostic platforms evaluated in this manuscript. By including it as a comparison point, the study aims to benchmark the performance of the experimental platforms, Biomeme Franklin™ and Truelab®, against a widely accepted and validated assay. The inclusion of the Cepheid platform strengthens the overall validity of the study by providing a well-established control for the comparison, reducing potential biases or doubts about the efficacy of the new assays (MolBio's Truelab® and Biomeme Franklin™). This also aligns with best practices in assay validation, where the performance of new diagnostic tools is often compared with those already proven to be accurate, ensuring that any differences in results are truly due to the platforms themselves and not to discrepancies in assay performance or methodology.

5- Figure 1, the name of SARS-CoV-2 genes should be added to the Cepheid SARS-CoV-2 and Truenat® COVID-19 assays.
As suggested the name of SARS-CoV-2 genes detected by the Cepheid SARS-CoV-2 and Truenat® COVID-19 assays has been added to all the panels of Figures 2 and 3.